# OpenReview forum: "Sharpness of Minima in Deep Matrix Factorization"
_ICML.cc/2026/Conference — Submitted to ICML 2026_

### Official Review · Reviewer_rtvN · 2026-03-10

**Soundness:** 4
**Presentation:** 3
**Significance:** 3
**Originality:** 4
**Overall Recommendation:** 5
**Confidence:** 3

**Summary:**

The paper studies the maximum eigenvalue of the Hessian of the squared-error loss as a characterization of local sharpness near minima in overparameterized deep matrix factorization.  The main contribution is an exact closed-form expression for the maximum Hessian eigenvalue at any global minimizer of the deep matrix factorization problem. This expression simplifies substantially in the deep overparameterized scalar factorization case and in the depth-2 matrix factorization case. Based on these formulas, the paper shows that for deep overparameterized scalar factorization, flatness is equivalent to a spectral-norm balancing condition across layers, and for depth-2 matrix factorization, flat minima are spectral-norm balanced but not necessarily Frobenius-norm balanced. The paper also discusses how different sharpness measures can lead to potentially misleading conclusions about flatness and balancedness. Additionally, the paper uses the derived sharpness formula to empirically characterize how models escape minimizers during training in deep matrix factorization problem.

**Compliance With Llm Reviewing Policy:**

Affirmed.

**Final Justification:**

The authors have mostly addressed my concerns, and I maintain my positive score.

**Key Questions For Authors:**

(1) A central conclusion is that flat minima are spectral-norm balanced, but not necessarily Frobenius-norm balanced, under the maximum Hessian eigenvalue criterion. Is spectral-norm balance the most informative notion in this setting? If so, can we expect that under different loss functions or problem settings, other notions of sharpness, including those more closely related to Frobenius-norm balance, may become appropriate?


(2) The current formulation focuses on overparameterized deep matrix factorization with squared-error loss. Is it possible to broaden the analysis beyond this setting, for example to more general deep linear models, in order to reduce the current limitation of the problem formulation?

**Limitations:**

Yes.

**Strengths And Weaknesses:**

**Strengths**

(1) The paper articulates clearly the role that the maximum eigenvalue of the Hessian plays in optimization, particularly through its connection to the local geometry of the loss landscape, providing a strong motivation.

(2) The paper makes strong statements and delivers sharp insights. It provides a concise characterization of the landscape around minima, and several of the conclusions are counter-intuitive yet well justified, with clear and detailed discussion.

(3) The experiments are well designed and supported by good reasoning. The empirical analysis is comprehensive, and the escape phenomenon is illustrated clearly.


**Weaknesses**

(1) The paper makes a strong contribution within deep matrix factorization, but the broader significance could be explained more clearly.

(2) The contribution is mainly in advancing the theory of deep matrix factorization and sharpness. Although this is valuable in its own right, the paper could do more to clarify the practical utility of the results, for example by discussing more explicitly what new understanding, or methodological guidance the exact characterization enables.

---

> ### Author Rebuttal · Authors · 2026-03-30
>
> We would like to thank the reviewer for their valuable questions and kind words regarding our work.
>
> **Response to Q1:** This question remains open in the literature. To this end, Josz (2025) defined the flat minima of any smooth function $\\mathcal{L}$ as the local minima of $\\lambda_{\\max}(\\nabla^2 \\mathcal{L}(\\mathbf{w}))$ under the constraint $\\mathcal{L}(\\mathbf{w}) = \\mathcal{L}(\\mathbf{w}^{\ast})$ and demonstrated that this notion coincides with other notions of flatness in the depth-2 matrix factorization.
>
> To the best of our knowledge, there is no robust sharpness measure, which reconciles existing interpretations of other sharpness measures, established yet for arbitrarily deep models or other problem settings for different loss functions. Therefore, we also do not know the answer of this question. We do, however, note that there has been a recent line of work linking the implicit bias of gradient descent to stable minima, where stability of a minimum is directly linked to the spectral norm of the Hessian of the loss (Wu et al., 2018; Mulayoff et al., 2021; Nacson et al., 2023). Thus, having a characterization of the spectral norm has been of recent theoretical interest.
>
> **Response to Q2:** The assumption that restricts the model architecture is as follows:
>
> $$
> \\min_{i} d_i \\geq \\min \\{d_0, d_L\\} \\quad \\forall i \\in [L].
> $$
>
> This condition ensures that any linear mapping from $\mathbb{R}^{d_0}$ to $\mathbb{R}^{d_L}$ can be expressed by the network. This is a natural corollary of the following:
>
> $$
> \\text{rank}\\left(\\mathbf{W}_L \\cdots \\mathbf{W}_1\\right) \\leq \\min \\left\\{\\text{rank}\\left(\\mathbf{W}_1\\right), \\dots, \\text{rank}\\left(\\mathbf{W}_L\\right)\\right\\}.
> $$
>
> The rank of each layer is upper bounded by its dimensions. Therefore, if the condition is violated, the column space of the end-to-end function cannot fill $\\mathbb{R}^{d_L}$. As a result, the optimal error is no longer $0$ at any global minimizer for any target matrix $\\mathbf{M}$. Therefore, without the assumption, our work holds only for target matrices $\\mathbf{M}$ that achieve zero training loss. To clarify this, **we will add the following subsection to Section 5:**
>
> > ***5.4. Limitations***
> >
> > *The dynamical stability analysis relies on how accurate the quadratic approximation of the loss function is in the $\\delta$-neighborhood of $\\mathbf{w}^{\\ast}$ (Wu et al., 2018) (also see Definition 1.1). For loss functions whose Hessian is degenerate at a minimizer, this analysis fails (Wu et al., 2018). For instance, Jules et al. (2023) observed that for a cross-entropy loss, the quadratic approximation of the training loss near a minimizer is fundamentally inadequate due to the exponential nature of the decision boundary. In contrast, the squared-error loss ensures that the Hessian at any minimizer is not degenerate/does not vanish (also see Mulayoff & Michaeli (2020, Lemma 3)). Therefore, our analysis cannot be extended to other loss functions for which their Hessian vanishes at their global minima. **Moreover, the condition required to guarantee the feasibility of factorization for any target matrix $\\mathbf{M}$, i.e. Equation (20), restricts the applicability of our work to more general settings. Without this condition, our analysis holds only for target matrices $\\mathbf{M}$ that achieve zero training error.***
>
> Jules, T., Brener, G., Kachman, T., Levi, N., and Bar-Sinai, Y. "Charting the topography of the neural network landscape with thermal-like noise." *arXiv preprint arXiv:2304.01335*, 2023

---

> > ### Author Rebuttal · Reviewer_rtvN · 2026-04-01
> >
> > Thank you for your detailed response, which clearly clarified the scope, assumptions, and limitations of the theory, and also broadened my understanding of this line of work. The authors have mostly addressed my concerns. I appreciate the thoughtful explanations, and I will maintain my positive score.

---

> > > ### Author Response · Authors · 2026-04-05
> > >
> > > We sincerely thank the reviewer for their insightful questions and kind words.

---

### Official Review · Reviewer_XEjX · 2026-03-11

**Soundness:** 3
**Presentation:** 2
**Significance:** 1
**Originality:** 2
**Overall Recommendation:** 3
**Confidence:** 3

**Summary:**

This paper investigates the geometric properties of loss terrain at global minima in Deep Matrix Factorization using squared error loss. The authors' main contribution is deriving the exact closed-form expression for the largest eigenvalue $\lambda_{max}$ of the Hessian matrix at any global minimum, refuting previous views that the problem is intractable. Based on this formula, the authors propose that a flat minima is equivalent to "spectral-norm balanced" between network layers, and point out that previous conclusions based on "Frobenius norm balanced" are misled by different sharpness metrics. Finally, by applying a perturbation along the direction of the largest eigenvector in the minimum neighborhood of the minimum, the authors verify the escape phenomenon of gradient descent (GD) when the learning rate $\eta > 2/\lambda_{max}$.

**Compliance With Llm Reviewing Policy:**

Affirmed.

**Final Justification:**

The rebuttal acknowledges that the theoretical framework assumes a setting where all minimizers generalize equally, which may not reflect practical scenarios in studying sharpness regularization. Additionally, the concern that rank-deficient Hessians are inherent to over-parameterized networks appears to remain insufficiently addressed. Consequently, the paper's contribution seems somewhat limited in its ability to explain practical deep learning phenomena. Given these unresolved limitations, I am inclined to maintain my weak reject.

**Key Questions For Authors:**

1. Could the authors please provide a rigorous mathematical definition of $\Omega_F$ (the set of flat minima) explicitly in the main text to avoid reader confusion?

2. Given a fully trained $w^*$, what are the specific, irreplaceable advantages of computing $\lambda_{max}$ via your closed-form solution compared to numerically estimating it using HVPs in standard autodiff frameworks? Why is the analytical solution strictly "necessary" for observing the escape phenomenon?

3. In Section 6, the authors state that if the perturbation is not orthogonal to the zero-eigenvalue space, GD "never converges to $w^\*$". In reality, it is necessary to distinguish whether the zero eigenvalue direction is a degenerate direction, such as x^4, which can still converge back to $w^\*$, or a truly degenerate direction without any gradient.

**Limitations:**

Although closed-form solutions are available for simple matrix factorization models, this theoretical approach is limited for general neural network sharpness research.

**Strengths And Weaknesses:**

Strengths

1. The closed-form solution of the Hessian maximum eigenvalue given in Theorem 3.1 and its corollaries is mathematically elegant and complete, filling a known gap in deep matrix factorization theory.

2. The article explicitly points out that using different sharness measures (such as Hessian trace and maximum eigenvalue) leads to drastically different inferences regarding the minimum equilibrium property (Frobenius vs. spectral norm). This provides a valuable warning and clarification for subsequent theoretical research in this field.

Weaknesses

1. The paper uses the notation $\Omega_F$ and flat minima directly and extensively in the main text (e.g., Corollaries 4.1 and 4.2), but fails to provide a clear mathematical definition anywhere in the body text. For a theoretically focused paper, the absence of such a key definition is a major oversight.

2. The authors attempt to verify the "escape phenomenon" in Section 6. However, the initial perturbation is set within a tiny $\delta$-neighborhood. At this microscopic scale, the quadratic approximation of the loss function is nearly perfect. In a strict quadratic terrain, it is a deterministic, classic optimization conclusion that the algorithm will inevitably diverge along the largest eigenvector if $\eta > 2/\lambda_{max}$. Therefore, the authors' claim that this "is the first empirical characterization of the escape phenomenon in deep matrix factorization" is overstated; these experiments merely confirm basic quadratic instability rather than revealing any nonlinear optimization dynamics unique to deep learning.

3. Theorem 3.1 provides a closed-form solution that requires the exact parameters at the minimum, $w^\*$, to be completely known. However, in practice, if a converged $w^*$ is already obtained, one can easily use Hessian-vector products combined with power iteration or the Lanczos algorithm to numerically approximate $\lambda_{max}$ with extreme efficiency. The authors claim this formula is "necessary" for experimentally verifying the escape phenomenon but fail to demonstrate the necessity of this analytical solution over standard numerical approximations.

4. The authors dedicate considerable space to deriving the closed-form solution but fail to leverage it to answer broader theoretical or practical questions. For instance, how does implicit regularization drive GD to find these "spectral-norm balanced" minima during training? How does "flat" minima give good generalization in the case of matrix factorization? The absence of such downstream exploration limits the paper's overall impact.

---

> ### Author Rebuttal · Authors · 2026-03-30
>
> We would like to thank the reviewer for their valuable insights. Since the first two questions are also covered in the Weaknesses section, we first reply to the concerns raised there, and then we answer Q3.
>
> *The paper uses the notation $\\Omega_F$ and flat minima directly...*
>
> **Response:** We would like to thank you for this remark. **We will define flat minima in Section 2** as follows:
>
> >$$
> >   \Omega_F := \arg\min_{\mathbf{w}^* \in \Omega}\lambda_{\max}\left(\nabla^2\mathcal{L}(\mathbf{w}^*)\right),
> >$$
> >*where flatness/sharpness is measured by the maximum Hessian eigenvalue. This sharpness measure is also known as worst-case sharpness.*
>
>
> *The authors attempt to verify the "escape phenomenon" in Section 6. However, the initial perturbation is set within a tiny -neighborhood...*
>
> **Response:** We agree on the fact that it is a conclusion from the classical optimization theory. Moreover, you can replace many other loss functions for which their Hessian does not vanish at their global minima and see the same effect. In the context of neural-network-like models, this phenomenon has been observed in numerous works; **however, it has been demonstrated empirically only in extremely simplified settings.** For instance, Lewkowycz et al. (2020, see Figure 1) observed this phenomenon for a one-hidden layer linear network trained with squared-error loss on a single scalar data point. Cohen et al. (2021) analyzed it for a convex quadratic loss function, and most recently, Ghosh et al. (2025, see Figure 4) characterized it empirically for the two-layer scalar factorization loss. We do not claim or state that we re-explored the escape phenomenon. **To address this, we will revise our statement about the escape phenomenon.**
>
> *Theorem 3.1 provides a closed-form solution that requires the exact parameters at the minimum, $w^{\\ast}$, to be completely known. However, in practice...*
>
> **Response:** We would like to thank you for this remark. We completely agree on the fact that an exact expression is not necessary to observe the escape phenomenon. **Accordingly, we will omit this statement from the manuscript.**
>
>
> *The authors dedicate considerable space to deriving the closed-form solution but fail to...*
>
> **Response:** The main purpose of our paper is to study the geometry of the loss landscape rather than training dynamics of gradient descent or the relationship between generalization and flatness. We leave the question of how this characterization impacts the training dynamics or the relationship between generalization and flatness in neural networks to future work. Furthermore, the deep matrix factorization/deep linear network training problem **is not well suited for analyzing the relationship between generalization and geometric properties of the landscape.** This is because deep linear network training problems with quadratic loss admit a unique end-to-end minimizing function determined by the second-order statistics of the training data (Mulayoff & Michaeli, 2020). **Consequently, any minimizer has exactly the same generalization ability, which makes such an analysis inapplicable in this setting.** Nonetheless, we do not think these limitations undermine our work, since linear networks are used extensively in the literature as toy models to understand the neural networks. (Mulayoff & Michaeli, 2020; Marion & Chizat, 2024; Ghosh et al., 2025). Furthermore, we would like to highlight that this closed-form expression shows that *flat minima are spectral-norm balanced in depth-2 matrix factorization, and they are not necessarily Frobenius-norm balanced*. We expect that this observation, together with the simple closed-form expression for the maximum Hessian eigenvalue of the depth-2 matrix factorization problem, will play an important role for future work on loss landscape geometry and the training dynamics of gradient descent, particularly due to the recent work linking the implicit bias of gradient descent to stable minima, where stability of a minimum is directly linked to the maximum Hessian eigenvalue of the loss (Wu et al., 2018; Mulayoff et al., 2021; Nacson et al., 2023).
>
> **Q3)** *In Section 6, the authors state that if the perturbation is not orthogonal to the zero-eigenvalue space, GD "never converges to $w^{\\ast}$". In reality...*
>
> **Response:** We would like to thank the reviewer for this remark. As we already mentioned in Section 6, our analysis fails for the loss functions whose Hessian is degenerate at its global minimizers. The example you gave, i.e. $x^4$, falls into this category. To clarify this, **we will add a subsection about limitations of our work to Section 5, which also discusses another limitation. See the Response to Reviewer L4Aw for the Limitations subsection.**
>
> Jules, T., Brener, G., Kachman, T., Levi, N., and Bar-Sinai, Y. "Charting the topography of the neural network landscape with thermal-like noise." *arXiv preprint arXiv:2304.01335*, 2023

---

> > ### Author Rebuttal · Reviewer_XEjX · 2026-04-01
> >
> > Thank you for the detailed rebuttal. While I appreciate the authors' transparency in acknowledging limitations, my core concerns remain unresolved.
> >
> > 1. Trivialized Generalization (W4): The authors state that in their setting "any minimizer has exactly the same generalization ability." Yet in practical tasks like Matrix Completion with insufficient data, flat vs. sharp minima exhibit drastically different generalization behaviors. If the theoretical framework cannot capture these differences, the motivation shown in the Introduction is severely undermined.
> >
> > 2. Hessian Degeneracy is a Necessity, Not an Edge Case (Q3): The authors admit their analysis fails for degenerate Hessians and propose adding a limitation. However, in over-parameterized models, global minimizers form a continuous zero-loss manifold — degeneracy is a strict mathematical necessity, not an edge case. The non-degeneracy assumption is too strong and limits applicability to modern neural network landscapes.
> >
> > 3. Diluted Core Claims (W2 & W3): Conceding that escape dynamics align with classical optimization theory, and that the closed-form solution is not actually necessary to observe them, this reduces the paper's primary contributions of Section 6.
> >
> > The closed-form expression is an elegant result, and I encourage the authors to build on it in future work targeting non-trivial settings (e.g., Matrix Completion) where these questions genuinely bite. That said, given the significant limitations and narrow scope of contribution, I can raise my score only marginally to 3 — no higher.

---

> > > ### Author Response · Authors · 2026-04-06
> > >
> > > We would like to thank the reviewer for updating their score.
> > >
> > > Although deep matrix factorization/deep linear neural network training problems are simplified models to study general deep neural architectures, they are strong toy models used to investigate phenomena induced by optimization concerns and the role of depth, regularization, and loss-landscape geometry of neural networks.
> > >
> > > *The authors admit their analysis fails for degenerate Hessians and propose adding a limitation. However, in over-parameterized models, global minimizers form a continuous zero-loss manifold — degeneracy is a strict mathematical necessity, not an edge case. The non-degeneracy assumption is too strong and limits applicability to modern neural network landscapes.*
> > >
> > > **Response:** We believe that there might be some confusion here, and we would like to clarify it. Yes, it is true that the loss landscape of deep matrix factorization problems (and general overparameterized problems) possesses continuous zero-loss manifolds, e.g., this is easy to see from the objective $\\mathcal{L}(x, y) := (xy-1)^2$. However, problems such as deep matrix factorization **do not** have degenerate Hessians at their minima (see Mulayoff & Michaeli, 2020). We would like to highlight that the terminology we adopted in our paper is that a degenerate matrix is one whose rank is $0$ (not merely rank deficient, as you are referring to).
> > >
> > > We would like to thank you for pointing out this unclear aspect of our exposition, and we will make this terminology clear with an explicit definition in the revision. Under our terminology, we cannot see the direct relationship between a continuous zero-loss manifold and the degeneracy of the Hessian at minima. This connection may not be straightforward. Furthermore, we are not aware of any references that prove that Hessian matrices of training losses of modern neural network architectures are rank $0$ at their minima. Could you provide some references for this statement? We would appreciate it.

---

### Official Review · Reviewer_x6Ao · 2026-03-12

**Soundness:** 3
**Presentation:** 3
**Significance:** 3
**Originality:** 3
**Overall Recommendation:** 4
**Confidence:** 3

**Summary:**

This paper studies the Hessian of deep linear networks at local minima, which are equivalently global minima. It computes a closed-form expression for the maximum eigenvalue of the Hessian at a minimizer. As a corollary, it shows that minima in deep linear networks are flat if and only if they satisfy a balancedness condition on the spectral norms of intermediate subnetworks. This contrasts the hypothesis that flat minima in deep networks arise from being Frobenius norm-balanced.

**Compliance With Llm Reviewing Policy:**

Affirmed.

**Final Justification:**

I maintain my initial positive evaluation of the work. The rebuttal helped clarify some of my questions and reinforced my initial assessment.

**Key Questions For Authors:**

See weaknesses.

**Limitations:**

yes

**Strengths And Weaknesses:**

## Strengths
- The paper is well-written and easy to follow. Definitions and proofs of theoretical results are rigorous.
- The sharpness of minima in deep linear networks is a fundamental question unanswered by prior work, and the main theorem characterizes it by finding an exact expression. In contrast to previous results which were restricted to particular subsets of parameter space, the result holds over the entire landscape.
- The discussion on spectral norm balancedness versus Frobenius norm balancedness improves intuition on the loss landscape of linear networks. The intuition from existing literature is that Frobenius norm balancedness over layers characterizes flat minima, and the authors point out that this only holds for minima reached by gradient descent under specific initializations, and are a special case of spectral norm balancedness.

## Weaknesses
- It is unclear whether the closed form expression for the Hessian eigenvalue is useful in and of itself. There is some discussion on how the sharpness of minima affects optimization as a principle, but limited discussion on how this particular result impacts optimization or generalization.
- I am not sure about what can be interpreted from the experiment. It does not seem specific to the setting of matrix factorization and seems to be demonstrating training dynamics of gradient descent at the edge of stability. Couldn't this be replaced by many other objective functions with the same effect?

---

> ### Author Rebuttal · Authors · 2026-03-30
>
> We would like to thank the reviewer for their valuable insights and kind words regarding our work. We address each concern raised in the Weaknesses section in turn.
>
> *It is unclear whether the closed form expression for the Hessian eigenvalue is useful in and of itself...*
>
> **Response:** The main purpose of our paper is to study the geometry of the landscape rather than training dynamics of gradient descent or the relationship between generalization and flatness. We leave the question of how this characterization impacts the training dynamics or the relationship between generalization and flatness in neural networks to future work. Furthermore, the deep matrix factorization/deep linear network training problem **is not well suited for analyzing the relationship between generalization and geometric properties of the landscape.** This is because deep linear network training problems with quadratic loss admit a unique end-to-end minimizing function determined by the second-order statistics of the training data (Mulayoff & Michaeli, 2020). **Consequently, any minimizer has exactly the same generalization ability, which makes such an analysis inapplicable in this setting.** Nonetheless, we do not think these limitations undermine our work, since linear networks are used extensively in the literature as toy models to understand neural networks. (Mulayoff & Michaeli, 2020; Marion & Chizat, 2024; Ghosh et al., 2025). Furthermore, we would like to highlight that this closed-form expression shows that *flat minima are spectral-norm balanced in depth-2 matrix factorization, and they are not necessarily Frobenius-norm balanced*. We expect that this observation, together with the simple closed-form expression for the maximum Hessian eigenvalue of the depth-2 matrix factorization problem, will play an important role for future work on loss landscape geometry and the training dynamics of gradient descent, particularly due to the recent work linking the implicit bias of gradient descent to stable minima, where stability of a minimum is directly linked to the maximum Hessian eigenvalue of the loss (Wu et al., 2018; Mulayoff et al., 2021; Nacson et al., 2023).
>
> *It does not seem specific to the setting of matrix factorization...*
>
> **Response:** Indeed, you can replace it with many other objective functions **for which their Hessian does not vanish at their global minima** and see the same effect. In the context of neural-network-like models, this phenomenon has been observed in numerous works; **however, it has been demonstrated empirically only in extremely simplified settings.** For instance, Lewkowycz et al. (2020, see Figure 1) observed this phenomenon for a one-hidden layer linear network trained with squared-error loss on a single scalar data point. Cohen et al. (2021) analyzed it for a convex quadratic loss function, and most recently, Ghosh et al. (2025, see Figure 4) characterized it empirically for the two-layer scalar factorization loss.

---

> > ### Author Rebuttal · Reviewer_x6Ao · 2026-04-03
> >
> > Thanks for your response and for addressing my questions. I will maintain my initial positive score.

---

> > > ### Author Response · Authors · 2026-04-05
> > >
> > > We thank the reviewer for their constructive feedback and positive score.

---

### Official Review · Reviewer_L4Aw · 2026-03-13

**Soundness:** 2
**Presentation:** 2
**Significance:** 3
**Originality:** 4
**Overall Recommendation:** 4
**Confidence:** 3

**Summary:**

The maximum Hessian eigenvalue is often used in the literature as a measure of flatness. The submission derives the expression for the maximum Hessian eigenvalue for any global minimum under the squared-error loss for deep matrix factorization. The preprint proceeds by considering special cases of deep scalar factorization and depth-2 matrix factorization, and by proving conditions on the spectral norms of the factors under which the minimum is flat. Finally, the submission provides the first empirical characterization of the minimum escape during gradient-based training for depth-2 matrix factorization and deep scalar factorization, using the derived expressions.

**Compliance With Llm Reviewing Policy:**

Affirmed.

**Final Justification:**

I find the derived exact expression for the maximum Hessian eigenvalue to be an interesting result and believe it can help further analysis of optimization dynamics and the escape phenomenon. Initially, the submission had several issues: the presentation was unclear in several places, and the experiment claims were not fully supported by the actual experimental results. Hence, the weaknesses slightly outweighed the strengths. However, the authors provided detailed explanations and proposed changes that fully addressed those concerns. Hence, I raised my score from Weak Reject to Weak Accept.

**Key Questions For Authors:**

**Crucial** (Questions affecting the score)

Q1. Presentation:

a) What is the exact mathematical definition of the minimizer being flat that is being used? Since flatness is generally not a binary property, an exact definition would help with clarity.

b) Can the definition of the norm-balanced minimizer be added to improve clarity?

c) Is it possible to add a more detailed explanation for Figure 1(c) and Figure 2? Specifically, how to interpret the changes in the plotted eigenvector values? Why do they usually start at 0, and why does $\mathbf{v}_N$ end at a non-zero value if there was an escape? Additionally, are $\mathbf{v}_1$ and $\mathbf{v}_N$ eigenvectors corresponding to the smallest and largest eigenvalues at different times, or are the smallest and largest eigenvalues measured at the start and then the corresponding eigenvectors tracked?

d) What are the work's limitations? For instance, what are the key assumptions? Can the results extend to other losses beyond squared-error loss or to nonlinear neural networks?

e) The Impact Statement is required by the submission guidelines.

Q2. Soundness:

The Abstract claims the empirical results on the escape phenomenon are for deep matrix factorization, but the results shown are for deep scalar factorization and depth-2 matrix factorization. While these are special cases of deep matrix factorization, is there a reason to consider them specifically as deep matrix factorization results? Can a similar experiment be run for depth-3 matrix factorization? The results are interesting, but the naming should be accurate.

**Minor** (Questions to confirm my understanding)

Q3. Does Corollary 4.4 use Theorem 3.1 in the proof, or can it be used to obtain the same result? Other results in Section 4 use corresponding results from Section 3, but this does not seem to be the case for deep matrix factorization.

**Limitations:**

See Key Questions For Authors.

**Strengths And Weaknesses:**

**Strengths**

S1.  Originality: Based on the related work overview, the submission is the first work to obtain the exact expression for the maximum Hessian eigenvalue in deep matrix factorization.

S2. Significance: The usefulness of this expression is clearly demonstrated by the derived characterization of the flat minima and the empirical characterization of the escape phenomenon. The results would be a valuable addition to the large body of work on understanding the geometry of the loss landscape.

S3. I appreciated the insightful discussion of the sharpness measures in Section 5, which aids in understanding the related work and the submission's contribution.

**Weaknesses**

W1. Presentation: While the writing is overall clear, there are several instances where clarity could be improved, mostly by adding definitions or an explanation. Additionally, the preprint would benefit from a discussion of the work limitations.

W2. Soundness: The claim about empirical results on the escape phenomenon in deep matrix factorization is not fully supported, as the results are for deep scalar factorization and depth-2 matrix factorization.

More details are in Key Questions For Authors.

Overall, the weaknesses currently slightly outweigh the strengths. However, they should be easy to address, and I would be happy to raise the score then.

---

> ### Author Rebuttal · Authors · 2026-03-30
>
> We would like to thank the reviewer for their valuable insights and kind words regarding our work. Since the concerns in the Weaknesses section are also covered in the Questions, we respond directly to the questions for the sake of conciseness.
>
> **Response to Q1.a:** We will define flat minima in Section 2 as follows:
>
> >$$
> >   \Omega_F := \arg\min_{\mathbf{w}^* \in \Omega}\lambda_{\max}\left(\nabla^2\mathcal{L}(\mathbf{w}^*)\right),
> >$$
> >*where flatness/sharpness is measured by the maximum Hessian eigenvalue. This sharpness measure is also known as worst-case sharpness.*
>
>
> **Response to Q1.b:** We will add **Definition 2.3** as follows:
>
> > *Since we use the notion of norm-balanced minima throughout the paper, we define it here for the sake of completeness.*
> >
> > **Definition 2.3.** A minimizer $\\mathbf{w}^{\\ast} \\in \\Omega$ is $\\ell^p$-norm balanced if
> >
> > $$\\left\\|\\mathbf{W}^{\\ast}\_1\\right\\|\_{\\mathcal{S}^p} = \\left\\|\\mathbf{W}^{\\ast}\_2\\right\\|\_{\\mathcal{S}^p} = \\cdots = \\left\\|\\mathbf{W}^{\\ast}\_L\\right\\|\_{\\mathcal{S}^p},$$
> >
> > *where $\\|\\cdot\\|\_{\\mathcal{S}^p}$ is the Schatten-$p$ norm. For instance, when $p=2$, the minimizer is Frobenius-norm balanced, and when $p = \\infty$, it is spectral-norm balanced.*
>
> **Response to Q1.c:** To observe the escape phenomenon, we need to chart the loss landscape around the minimizer $\\mathbf{w}^{\\ast}$; however, the parameter space is high-dimensional, and this prevents a full visualization. Therefore, the common practice is to project the loss landscape onto two vectors to chart a 2D contour plot (Goodfellow et al., 2014; Li et al., 2018). We select these vectors as eigenvectors of $\\nabla^2 \\mathcal{L}(\\mathbf{w}^{\\ast})$ that correspond to the maximum and minimum eigenvalues of  $\\nabla^2 \\mathcal{L}(\\mathbf{w}^{\\ast})$. To track the training dynamics in the projected space, we project the iterates onto the subspace spanned by these two eigenvectors and indicate each iterate on the 2D contour plot. Furthermore, the initialization is in a $\\delta$-neighborhood of $\\mathbf{w}^{\\ast}$ as we explained in Section 6. Therefore, the dynamics start arbitrarily close to the origin in the projected space. Therefore, if the training dynamics converge to a point that is far away from the origin relative to the initialization, we can conclude that the escape phenomenon occurred. Moreover, we expect the oscillations stemming from the instability of the minimizer for a given learning rate in the direction of the eigenvector that corresponds to the largest eigenvalue  (Cohen et al., 2021).
>
> **According to the points above, we will revise the Section 6 and Appendix D.2 substantially. To clarify the figures, we will revise the captions of Figure 1 and Figure 2.**
>
> **Response to Q1.d:** We would like to thank you for this remark. As noted in the Weaknesses section, the manuscript lacks a limitations section. Hence, **we will add the following subsection to Section 5:**
>
> > ***5.4. Limitations***
> >
> > *The dynamical stability analysis relies on how accurate the quadratic approximation of the loss function is in the $\\delta$-neighborhood of $\\mathbf{w}^{\\ast}$ (Wu et al., 2018) (see Definition 1.1). For loss functions whose Hessian is degenerate at a minimizer, this analysis fails (Wu et al., 2018). For instance, Jules et al. (2023) observed that for a cross-entropy loss, the quadratic approximation of the training loss near a minimizer is fundamentally inadequate due to the exponential nature of the decision boundary. In contrast, the squared-error loss ensures that the Hessian at any minimizer is not degenerate/does not vanish (also see Mulayoff & Michaeli (2020, Lemma 3)). Therefore, our analysis cannot be extended to other loss functions for which their Hessian vanishes at their global minima. Moreover, the condition required to guarantee the feasibility of factorization for any target matrix $\\mathbf{M}$, i.e. Equation (20), restricts the applicability of our work to more general settings. Without this condition, our analysis holds only for target matrices $\\mathbf{M}$ that achieve zero training error.*
>
> **Response to Q1.e:** We would like to thank you for this remark. We will include an impact statement in the revised manuscript.
>
> **Response to Q2:** We would like to thank you for this remark. To that end, we will extend our experiments to cover depth-$3$ matrix factorization in the revised manuscript.
>
> **Response to Q3:** Corollary 4.4 uses the inequality in (37), which is derived in Appendix B (see Equation (69)). Nonetheless, this inequality can also be derived directly from Theorem 3.1 by using the triangle inequality and the fact that $\\sigma_{\\max}(\\mathbf{A} \\otimes \\mathbf{B}) =\\sigma_{\\max}(\\mathbf{A})\\sigma_{\\max}(\\mathbf{B})$.
>
>
> Jules, T., Brener, G., Kachman, T., Levi, N., and Bar-Sinai, Y. "Charting the topography of the neural network landscape with thermal-like noise." *arXiv:2304.01335*, 2023

---

> > ### Author Rebuttal · Reviewer_L4Aw · 2026-04-04
> >
> > I thank the authors for the detailed response and proposed changes. All issues I raised have been resolved. I especially appreciate the added experiment addressing soundness, and the proposed definitions and explanations make the results much clearer.
> >
> > I find the derived exact expression for the maximum Hessian eigenvalue to be an interesting result and believe it can help further analysis of optimization dynamics and the escape phenomenon. I will raise my score from Weak Reject to Weak Accept.

---

> > > ### Author Response · Authors · 2026-04-05
> > >
> > > We sincerely thank the reviewer for the detailed feedback and for updating their score.

---

### Decision · Program_Chairs · 2026-04-30

**Decision:**

Reject

**Comment:**

This paper derives a closed-form expression for the maximum Hessian eigenvalue at arbitrary global minimizers of the squared-error deep matrix factorization (DMF) objective. It uses this expression to characterize flat minima via spectral-norm balancing, contrasting this with the Frobenius-norm balancing that appears under other sharpness measures, and to study escape behavior near minimizers in depth-2 matrix factorization and deep scalar factorization.

**Strengths.** Theorem 3.1 is a real technical contribution: it gives the first exact closed-form expression for $\lambda_{\max}$ at arbitrary global minima of overparameterized DMF, resolving a question explicitly flagged by Mulayoff & Michaeli (ICML 2020) as intractable. Reviewers also valued the clarification between spectral- and Frobenius-based notions of flatness. The rebuttal materially improved the paper by adding a depth-3 experiment, clarifying definitions, adding a limitations subsection, and scaling back the original claims in Section 6; on this basis L4Aw raised $3 \to 4$ and XEjX raised $2 \to 3$.

**Weaknesses.** A limitation remains in the scope of the result. As the authors acknowledge in rebuttal, in the analyzed regime all minimizers have the same generalization behavior, so the main result is a landscape-geometry result rather than a generalization result. In addition, Section 6 appears to verify the classical local-instability condition $\eta > 2/\lambda_{\max}$ rather than establishing a distinctly DMF-specific phenomenon; the authors agree and commit to revising this framing.

**AC analysis.** In my reading, the paper studies sharpness for a fixed end-to-end solution in a deterministic matrix-matching setting, where by construction every zero-loss minimizer implements the same end-to-end linear map. The more scientifically consequential question — which end-to-end solutions are preferred by sharpness minimization in underdetermined settings, and whether that induced bias helps explain generalization — is not answered within the analyzed problem. I see this as limiting the paper's impact, but not enough to outweigh the value of the theorem itself. A particularly promising extension would be to push this line toward underdetermined recovery or completion settings, in the spirit of recent work by Ding et al. and Gatmiry et al., where parameter-space flatness can induce a nontrivial end-to-end bias. That would connect the present closed-form landscape result more directly to the broader questions of implicit bias and generalization that motivate the paper.

**Decision.** The paper is **borderline**. The paper presents a clean and technically nontrivial closed-form result that resolves an explicit open problem from Mulayoff & Michaeli (ICML 2020). While Reviewer XEjX correctly points out that the current setting does not engage the more consequential sharpness-vs-generalization question, the majority of the reviewers and I agree that the derivation of Theorem 3.1 and the clarification between spectral- and Frobenius-based sharpness notions offer sufficient value to the community. For the camera-ready version, the authors should tighten the abstract and introduction so that the delivered contribution is scoped explicitly as a landscape-geometry theorem for deep matrix factorization, and should incorporate the promised revisions to Section 6.